# Subgroup Characteristics of Middle-Aged and Older Women with Chronic Low Back Pain by Multiple Factors: A Hierarchical Cluster Analysis

**DOI:** 10.3390/jfmk10010030

**Published:** 2025-01-14

**Authors:** Ryo Miyachi, Takaaki Nishimura, Masahiro Noguchi, Akio Goda, Hiromichi Takeda, Eisuke Takeshima, Yuji Kanazawa, Tadashi Imai, Wataru Tanaka

**Affiliations:** 1Faculty of Health and Medical Sciences, Hokuriku University, 1-1 Taiyogaoka, Kanazawa 920-1180, Japan; t-nishimura@hokuriku-u.ac.jp (T.N.); m-noguchi@hokuriku-u.ac.jp (M.N.); a-goda@hokuriku-u.ac.jp (A.G.); h-takeda@hokuriku-u.ac.jp (H.T.); e-takeshima@hokuriku-u.ac.jp (E.T.); yu_kanazawa@hokuriku-u.ac.jp (Y.K.); 2Rehabilitation Center, Kanazawa Nishi Hospital, 6-15-41 Ekinishihonmachi, Kanazawa 920-0025, Japan; pt101imai@gmail.com; 3Department of Rehabilitation, Komatsu Sophia Hospital, 478 Okimachi, Komatsu 923-0861, Japan; wataru.sophia.hosp@gmail.com

**Keywords:** muscle mass, movement control, autonomic balance, tenderness threshold, central sensitization

## Abstract

Background/Objectives: Chronic low back pain (CLBP) after middle age is a complex multifactorial condition, and subgrouping is recommended to determine effective treatment strategies. Multidimensional data help create new groupings to increase the effectiveness of interventions in middle-aged and older adults with CLBP. This study aimed to investigate the relationship between the factors associated with CLBP after middle age and to create and characterize a new subgroup based on these factors. Methods: A cross-sectional observational study was conducted and included 46 women aged ≥40 years with CLBP who participated in health events. Trunk muscle mass, lumbar movement control ability, autonomic balance, lumbar tenderness threshold, lumbar proprioception, and severity of central sensitization were assessed. Results: Partial correlation analysis revealed a significant negative correlation between lumbar movement control ability and autonomic balance. A significant positive correlation was observed between trunk muscle mass and the lumbar tenderness threshold. Hierarchical clustering analysis identified three subgroups. The cluster 1 participants had low trunk muscle mass, low tenderness threshold, and low severity of central sensitization. The cluster 2 participants had low trunk muscle mass and tenderness threshold and high severity of central sensitization. The cluster 3 participants had high trunk muscle mass and tenderness threshold and were sympathetically predominant. Trunk muscle mass, pressure pain threshold, severity of central sensitization, and autonomic balance were significantly different between the clusters. Conclusions: Three characteristic subgroups were identified. The results contribute to treatment and prevention strategies for middle-aged and older adults with CLBP based on the characteristics of the subgroups rather than a uniform approach.

## 1. Introduction

Low back pain is one of the most common health problems with a high lifetime prevalence of 84% and a trend toward longer years lived with disability [1,2]. In particular, chronic low back pain (CLBP) not only reduces the quality of life at the individual level through pain and associated activity limitations, hospital visits, and restrictions on work and hobbies but also causes significant socioeconomic losses due to increased medical costs and decreased labor productivity [1,2,3]. The prevalence of CLBP increases around the age of 40 years, when age-related changes begin to occur, and CLBP after middle age is considered a worldwide problem [3,4]. Despite the high prevalence of CLBP in middle-aged and older individuals, many aspects remain unclear due to the involvement of various factors associated with aging-related changes. Therefore, evidence is needed for the prevention and treatment of CLBP after middle age.

In recent years, CLBP has been viewed as a complex condition involving multiple factors, requiring multifaceted evaluation and treatment [5]. Many studies have investigated low back pain in terms of biomechanics, such as trunk muscle size and lumbar movement control (LMC) ability [5,6,7,8,9,10]. However, CLBP is related not only to biomechanical factors but also to various other factors, such as the proprioceptive system, autonomic nervous system, central nervous system, and lifestyle [11,12,13,14,15]. Previous studies have shown that individuals with CLBP tend to have poorer proprioception, increased sympathetic dominance, heightened central sensitization, and less healthy lifestyle habits compared to those without CLBP [11,12,13,14,15]. Therefore, comprehensive evaluation and treatment from multiple perspectives, rather than limitation to biomechanical factors, are important for preventing and improving CLBP.

To provide efficient and effective interventions for CLBP, it is recommended to identify subgroups that share some characteristics [16]. Low back pain is a general term for pain in the lumbar region (the area on the posterior surface of the body from the lower margin of the 12th rib to the lower buttocks) [3]. Because not all patients with CLBP have the same conditions and factors, it is difficult to establish a uniform treatment method for CLBP. This has led to the subgrouping of CLBP, and matching treatments with subgroups has begun to yield good results [16]. Subgrouping has been conducted from various perspectives, including direction of motion related to symptoms, mechanism of injury, kinematic parameters of the lumbopelvic girdle, fear of movement, and magnitude of risk [8,16,17,18]. Despite the importance of multidimensional perspectives in CLBP, most studies classify subgroups based on limited factors rather than using multidimensional data [19,20]. The limited use of multidimensional data in previous studies may be attributed to the diversity of factors associated with CLBP and the complexity of their interactions [19]. Focusing on clinically important and modifiable factors and establishing multidimensional subgroups is expected to contribute to the development of novel classifications that enhance the effectiveness of interventions for middle-aged and older adults with CLBP.

Therefore, this study aimed to investigate the relationship between several factors that may be associated with CLBP after middle age and to create and characterize a new subgroup based on these factors.

## 2. Materials and Methods

### 2.1. Study Design and Participants

This was a cross-sectional observational study conducted at a community health event held in September 2024, and participants were asked to cooperate in the study. All measurements were performed indoors at a local health facility. The participants were aged ≥40 years; middle-aged and older individuals with CLBP (back pain lasting >3 months) were included in the study as age-related changes become apparent around 40 years of age [21]. Additionally, most of the participants in the health events were women, which limited the scope of this study to women. Exclusion criteria were spondylitis, large disc herniation/severe slip/spinal canal stenosis with obvious neurological symptoms, previous surgery that significantly affected spinal motion, such as spinal fusion, typical physical function impairment, such as paralysis due to cerebrovascular disease, inability to understand the study content due to cognitive impairment, pregnancy, and difficulty in performing the measurement positions in this study (standing without upper limb support, prone, or crawling). Of 154 community health event participants, 46 (age: 67.9 [9.8] years, height: 154.0 [5.2] cm, and weight: 50.6 [9.9] kg) who met the above inclusion criteria and did not meet the exclusion criteria were included (Figure 1).

In accordance with the Declaration of Helsinki, the purpose and content of the study, the fact that the data obtained would not be used for any purpose other than the study, and the precautions against leakage of personal information were fully explained to the participants in advance as ethical considerations. Consent for participation was obtained before the study was conducted. This study was approved by the ethics committee of Hokuriku University (approval number: 2024-1).

### 2.2. Measurement

Measurements included trunk muscle mass, LMC ability, autonomic balance, lumbar pressure pain threshold (LPPT), proprioception of the lumbar region, and severity of central sensitization.

Trunk muscle mass was measured using bioelectrical impedance analysis with a body component analyzer (Inbody 270, Inbody Japan Inc., Tokyo, Japan). The participants stood barefoot on the plantar sensor of the device and grasped the upper limb sensor during measurement.

The LMC test battery was used to evaluate LMC ability. The LMC test battery comprised six tests (waiters bow, pelvic tilt, one leg stance, sitting knee extension, transfer of the pelvis forwards and back in a quadruped position, and prone lying active knee flexion), with one point added for each positive test (poor motor control), and a score between 0 and 6. The evaluators were physical therapists with at least 10 years of experience, who were trained in the assessment of motor control. Criteria for positive results were discussed among the coauthors in advance and practiced to ensure consistent evaluation criteria. During testing, the participants were given standardized instructions that were similar to those used in previous studies [22].

Autonomic balance was assessed using a pulse analyzer (TAS9 VIEW, YKC, Tokyo, Japan) to measure the acceleration pulse waves from the index finger. The measurements were taken in a sitting position with eyes closed in a restricted space where no information from the outside could be received. A frequency-domain power spectrum density analysis was performed on the obtained data to extract the heart rate variability (HRV) spectral components: low-frequency (LF: 0.04–0.15 Hz) and high-frequency (HF: 0.15–0.4 Hz) frequency HRV were extracted [23]. HF HRV is generally considered to reflect parasympathetic activity, and LF HRV is often used as an indicator of sympathetic activity [24]. In this study, the LF/HF ratio was calculated as an indicator of the sympathetic and parasympathetic balance. Although a measurement time of 3 min is recommended for LF and HF [25], the Bland–Altman analysis confirmed that there was no systematic error between the 3 and 1 min values using the equipment employed in this study.

The LPPT of the lumbar erector spinae muscles is often used as an indicator of hyperalgesia [26] and was measured using a pressure measuring instrument (AMF Digital Force Gauge AMF-300, YAHUJI, Tokyo, Japan) with a 1 cm diameter probe. Pressure was applied perpendicularly to the tissue surface thrice at a constant rate (1 kgf/s). Pressure was applied to the right longissimus muscle belly at the level of the fourth lumbar vertebra. The LPPT was measured by instructing the participants to say ‘stop’ when the pressure sensation changed to “uncomfortable” or “painful”; the average of three measurements was taken as the representative value [26].

The joint repositioning test with lumbar flexion in the sitting position was used to measure lumbar proprioception [27]. The lumbar flexion angle was defined as the tilt angle of the thoracolumbar transitional area relative to the pelvis and was evaluated using the iPhone^®^ inclinometer application, as in a previous study [28]. The starting position was with eyes closed and the upper limbs crossed in front of the chest at a lumbar flexion angle of 0°. The participants flexed the lumbar spine in a voluntary movement with the pelvis fixed, and after memorizing the position where the lumbar spine flexion angle was 20°, they performed the trunk flexion movement again in a voluntary movement and stopped at the position where they felt the lumbar spine moved 20°. The measurements were performed three times, and absolute errors were calculated from the obtained measurements [29].

A questionnaire was used to obtain basic information on the location and degree of pain (numerical rating scale (NRS)), medical and surgical history, and severity of central sensitization. The central sensitization inventory-9 (CSI), a shortened version of the central sensitization inventory, was used as an indicator of the severity of central sensitization [30].

### 2.3. Statistical Analysis

Statistical analysis was performed using R version 4.4.2 and SPSS version 28 (IBM SPSS Statistics, IBM Japan, Tokyo, Japan). After confirming normality using the Shapiro–Wilk test (Holm correction), a partial correlation analysis was conducted using age as a control variable to verify the relationships among the items. Hierarchical cluster analysis using the Ward method with Euclidean distance, which is suitable for small sample sizes, was conducted to classify the subgroups based on each item [31,32]. A z-score was used for the cluster analysis of each variable. The optimal number of clusters was determined using a dendrogram. To confirm the validity of the clusters, differences between clusters were compared. One-way analysis of variance and multiple comparisons (Turkey correction) were performed for items for which normality was confirmed by the Shapiro–Wilk test, and the Kruskal–Wallis test was performed for items for which normality was not confirmed to compare variables between clusters. The significance level for the statistical analysis was set at 0.05.

## 3. Results

The participant characteristics and CLBP-related items are shown in Table 1. The LMC test scores and NRS did not confirm normality. The correlation coefficients for the CLBP-related items are shown in Table 2. A significant negative correlation was observed between the LMC test score and the LF/HF ratio (*p* = 0.01 and r = 0.37). Additionally, there was also a significant positive correlation between the trunk muscle mass and LPPT (*p* = 0.01 and r = 0.37). No significant correlations were observed between the other parameters.

Cluster analysis classified the participants into three groups: cluster 1, low trunk muscle mass and LPPT, good lumbar proprioception, and low severity of central sensitization; cluster 2, low trunk muscle mass and LPPT, no sympathetic dominance, and high severity of central sensitization; and cluster 3, high trunk muscle mass and LPPT, sympathetic dominance, and low severity of central sensitization.

Multiple comparisons revealed no significant differences in basic characteristics, such as age, body mass index, or NRS among the clusters. The CSI score for cluster 2 was significantly higher than that for cluster 1 (*p* < 0.01 and r = 0.70). Between clusters 1 and 3, trunk muscle mass (*p* < 0.01 and r = 0.61) and LPPT (*p* < 0.01 and r = 0.55) in cluster 3 were significantly greater than those in cluster 1. Between clusters 2 and 3, trunk muscle mass (*p* < 0.01 and r = 0.71), LF/HF ratio (*p* = 0.02 and r = 0.50), and LPPT (*p* < 0.01 and r = 0.83) were significantly greater in cluster 3 than those in cluster 2. The CSI of cluster 2 was significantly higher than that of cluster 3 (*p* = 0.04 and r = 0.40) (Table 3). Although there were no significant differences in absolute error, moderate to large effect sizes [33] were found, with cluster 3 being higher than clusters 1 (*p* = 0.06 and r = 0.51) and 2 (*p* = 0.10 and r = 0.42).

## 4. Discussion

This study investigated the relationships between several factors that could be related to CLBP in middle-aged and older adults (trunk muscle mass, LMC ability, autonomic balance, LPPT, lumbar proprioception, and severity of central sensitization) and the differences in the characteristics of each subgroup based on these factors.

In this study, significant correlations were identified between the LMC test score and autonomic balance (r = −0.37) and between trunk muscle mass and LPPT (r = 0.37), both demonstrating a moderate magnitude [33]. These results provide important preliminary insights into the potential relationships among the multidimensional parameters associated with CLBP. The observed correlation magnitudes suggest that CLBP is not solely influenced by a single factor, but rather by the combined effects of multiple interacting factors.

The present results show a significant negative correlation between the LMC test score and autonomic balance, suggesting that a higher LMC test score is associated with higher sympathetic or lower parasympathetic activity. Previous studies have reported that when sympathetic dominance occurs due to pain or other stressors, muscle activity increases, global co-contraction is prioritized, proprioceptive input decreases, and fine movement control is impaired [32]. This suggests that sympathetic dominance is associated with a “stiffening” movement control strategy rather than a “flexible” one. In fact, it has frequently been reported that individuals with CLBP exhibit higher LF HRV dominance than HF HRV and reduced lumbar movement compared to healthy individuals [9,34]. Among the six tasks included in the movement control test used in this study, five are designed to reduce lumbar movement through simultaneous contraction of the trunk muscles. Therefore, it should be noted that a “high LMC ability” as assessed by the LMC test score in this study reflects a proficiency in reducing lumbar movement. This may explain why greater sympathetic activity was associated with better LMC ability in the results of this study.

Regarding the positive correlation between trunk muscle mass and LPPT in the present results, a previous study reported a positive correlation between LPPT and the cross-sectional area of the trunk muscles [26], and the present study supports this finding. Although the causal relationship between pressure pain sensitivity and muscle size in patients with CLBP remains unclear, inflammatory mediators and peripheral inflammatory processes are thought to be involved [26]. It has also been verified that individuals with CLBP have a higher sensitivity to pressure pain [26,35,36]. The back muscles of the lumbar region become atrophied when CLBP occurs [5], and pressure pain sensitivity may be higher in patients with reduced trunk muscle mass due to the severity of CLBP. Additionally, higher physical activity is associated with lower pressure pain sensitivity [35], and trunk muscle mass may be a mediating factor for physical activity, leading to the present results. Furthermore, in the present study, pressure pain sensitivity was not simply measured in the muscles alone but was considered to include deep tissues. Therefore, it is possible that in muscles with a small size and poor elasticity, the muscle could not act as a cushion for the deep tissues, resulting in a lower tenderness threshold. However, the present study lacks information that would provide a clear explanation of the mechanism underlying the results, and further verification, including other factors, such as the amount of physical activity, is needed.

Considering the results of multiple comparisons of the characteristics of each subgroup, cluster 1 had low trunk muscle mass, LPPT, and low severity of central sensitization. Cluster 2 was a subgroup with low trunk muscle mass and LPPT, low sympathetic nerve activity, and high-severity central sensitization. Cluster 3 was a subgroup with high trunk muscle mass, LPPT, sympathetic activity, and low lumbar proprioception. The reason for no significant difference in lumbar proprioception could be due to sample size, and this result is consistent with that of previous studies [37], which showed that proprioception-based control is less likely to occur when the sympathetic nervous system is dominant.

The results of this study revealed the existence of subgroups based on multifaceted factors and highlighted their characteristics. These results suggest the need for treatment and prevention tailored to subgroup characteristics, even in interventions for populations, rather than uniform interventions for middle-aged and older adults with CLBP. The subgroups characterized in this study may offer more effective intervention strategies. Although there are commonalities among the subgroups, cluster 1, for example, may need to focus on training to increase trunk muscle mass, while cluster 2 may need to focus not only on increasing trunk muscle mass but also on interventions for central sensitization (such as pain neuroscience education, cognitive behavioral therapy, and multimodal lifestyle interventions, including physical activity, sleep, nutrition, and stress management) [38,39,40,41,42]. Cluster 3 may need to prioritize treatments related to sympathetic inhibition [43,44], such as breathing exercises, mindfulness meditation, graded motor imagery, sensorimotor treatment, and mirror visual feedback, and fine lumbar motor control exercises based on lumbar proprioception. However, it is unclear whether the characteristics of the subgroups are factors in CLBP, and further validation, including response to each treatment, is needed. Moreover, in this study, subgroups were identified based on key factors; however, individual variability within subgroups may affect the generalizability of the results. Therefore, further validation with larger sample sizes and evaluations of the reproducibility of the clustering results are necessary to ensure the generalizability of the clusters for clinical application. Additionally, although subgrouping may streamline the evaluation process by reducing the number of assessment items, the complexity of evaluating the specific characteristics of each subgroup cannot be avoided. Furthermore, the lack of clear criteria for grouping highlights the need to establish evaluation protocols, including reference values, and to reconsider variables that can distinctly separate groups for future clinical application of these findings.

### Limitations

This study was conducted with participants who attended a health event; therefore, it is possible that the participants were health-conscious and had good motor functions. Different results may be obtained for those with a level of motor or mental function that prevents them from leaving their homes, or for those with reduced activity levels. This limitation is also reflected in the exclusion of participants in this study who had difficulty assuming the required measurement position. Additionally, only women were included in this analysis. Muscle mass and mental status may differ between men and women, and factors related to CLBP may also differ. Therefore, the results of this study need to be interpreted by considering the participant characteristics, and validation in different populations is necessary. Additionally, the LMC test in this study included various elements, and it has been reported that each element of LMC is independent [45]; future verification of each element of LMC ability may be necessary. In evaluating autonomic balance, this study adopted a 1 min measurement after confirming the absence of systematic error using the Bland−Altman analysis. However, despite differences in devices and measurement sites, the measurement duration is shorter than the 3 min duration recommended in previous studies [25], and the potential effect of this shortened duration on the results cannot be entirely excluded. Additionally, although this study utilized the LF/HF ratio as an indicator of autonomic balance, future studies should consider incorporating other parameters, such as absolute values like total power [25], which may require longer measurement durations for accurate evaluation. Furthermore, this study did not collect data on lifestyle factors such as physical activity, sleep, and diet, which have been suggested to be associated with low back pain [7,15], and these factors were not considered. Future studies should incorporate these lifestyle factors to enable a more comprehensive analysis.

## 5. Conclusions

This study investigated the relationship among several factors that could be involved in CLBP (trunk muscle mass, LMC ability, autonomic balance, LPPT, lumbar proprioception, and severity of central sensitization) in middle-aged and older women. As a result, correlations were found between trunk muscle mass and LPPT and between autonomic balance and LMC ability. Additionally, three characteristic subgroups were identified. These results may contribute to treatment and prevention strategies tailored to the characteristics of the subgroups, rather than one-size-fits-all interventions.

## Figures and Tables

**Figure 1 jfmk-10-00030-f001:**
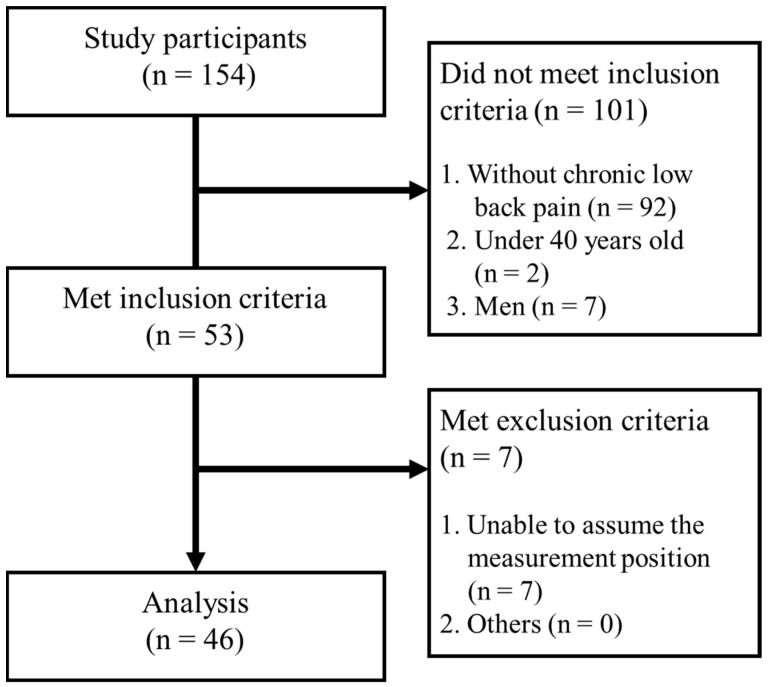
Flowchart of the study participants.

**Table 1 jfmk-10-00030-t001:** General characteristics and measurements of participants.

	Overall (n = 46)	Cluster 1 (n = 13)	Cluster 2 (n = 24)	Cluster 3 (n = 9)
Age (years)	67.9 (9.8)	71.5 (11.6)	66.0 (9.5)	68.0 (7.1)
BMI (kg/m^2^)	21.3 (3.7)	21.0 (3.2)	20.7 (3.3)	23.5 (5.0)
NRS (score)	4.4 (2.2)	4.8 (2.7)	4.6 (2.0)	3.6 (1.9)
Trunk muscle mass (kg)	15.1 (2.3)	14.2 (1.8)	14.6 (1.5)	17.8 (3.0)
LF/HF ratio	1.4 (2.8)	1.6 (2.2)	0.6 (0.5)	3.5 (5.3)
LMC test (score)	2.8 (1.2)	3.1 (1.2)	2.9 (1.0)	2.3 (1.7)
LPPT (kgf)	9.5 (4.4)	9.2 (4.3)	7.6 (2.1)	15.0 (4.7)
AE (degree)	3.0 (2.5)	2.3 (1.2)	2.8 (2.0)	4.7 (4.0)
CSI (score)	11.4 (5.9)	6.2 (4.0)	14.8 (4.5)	10.1 (5.9)

Values are presented as mean (standard deviation). BMI, body mass index; NRS, numerical rating scale; LF, low frequency; HF, high frequency; LMC, lumbar movement control; LPPT, lumbar pressure pain threshold; AE, absolute error of the joint repositioning test; CSI, central sensitization inventory.

**Table 2 jfmk-10-00030-t002:** Correlation coefficient for each measurement item.

	LF/HF Ratio		LMC Test		LPPT		AE		CSI	
	*p*-Value	Correlation Coefficient (r)	*p*-Value	Correlation Coefficient (r)	*p*-Value	Correlation Coefficient (r)	*p*-Value	Correlation Coefficient (r)	*p*-Value	Correlation Coefficient (r)
Trunk muscle mass	0.71	0.06	0.54	−0.09	0.01 *	0.37	0.49	0.11	0.57	−0.09
LF/HF ratio			0.01 *	−0.37	0.33	0.15	0.65	0.07	0.64	0.07
LMC test					0.69	0.06	0.59	0.08	0.83	−0.03
LPPT							0.43	0.12	0.70	0.06
AE									0.58	−0.09

* Significant correlation (*p* < 0.05). LF, low frequency; HF, high frequency; LMC, lumbar movement control; LPPT, lumbar pressure pain threshold; AE, absolute error; CSI, central sensitization inventory.

**Table 3 jfmk-10-00030-t003:** Group comparison for each cluster.

	Cluster 1 vs. Cluster 2		Cluster 1 vs. Cluster 3		Cluster 2 vs. Cluster 3	
	*p*-Value	Effect Size (r)	*p*-Value	Effect Size (r)	*p*-Value	Effect Size (r)
Age (years)	0.24	0.25	0.69	0.18	0.85	0.15
BMI (kg/m^2^)	0.95	0.06	0.28	0.30	0.13	0.06
NRS (score)	1.00	0.03	0.89	0.20	1.00	0.14
Trunk muscle mass (kg)	0.87	0.10	<0.01 *	0.61	<0.01 *	0.71
LF/HF ratio	0.50	0.42	0.21	0.25	0.02 *	0.5
LMC test (score)	1.00	0.05	0.63	0.25	0.80	0.19
LPPT (kgf)	0.37	0.31	<0.01 *	0.55	<0.01 *	0.83
AE (degree)	0.82	0.15	0.06	0.51	0.10	0.42
CSI (score)	<0.01 *	0.70	0.13	0.39	0.04 *	0.4

* Significant difference (*p* < 0.05). BMI, body mass index; NRS, numerical rating scale; LF, low frequency; HF, high frequency; LMC, lumbar movement control; LPPT, lumbar pressure pain threshold; AE, absolute error; CSI, central sensitization inventory.

## Data Availability

The data are available upon reasonable request from the corresponding author.

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
