# Peer review of "Subgroup Characteristics of Middle-Aged and Older Women with Chronic Low Back Pain by Multiple Factors: A Hierarchical Cluster Analysis"

_jfmk, 2025, doi:10.3390/jfmk10010030_

Round 1
Reviewer 1 Report
Comments and Suggestions for Authors
- This article examines the relationships between various factors affecting back pain for ways to improve the condition of patients;
- In my opinion, this topic is relevant. When we can understand the mechanisms of back pain, we will be able to develop methods for preventing and improving this pathology;
- This study adds an algorithm to pain development and management options;
- The results and conclusions are related to the aim of the article and I believe that they describe the back pain process. This is important for future prevention;
- I think the references are very appropriate; - Figure 1 not entirely clear.
It is necessary to add in conclusion about the connections with sympathetic activity and the capacity to control lumbar movements (LMC). The negative correlation between LMC and LF/HF is not only a connection of proprioceptive mechanisms of movement control. An increase in the autonomic balance LF/HF indicates a strain of autonomic regulation due to stress associated with pain syndrome. It is this mechanism that weakens the patient's motor control.
Author Response
Thank you for pointing this out. We have revised it as follows.
Point 1: Figure 1 not entirely clear.
Response 1:
We have corrected it as you suggested.
[P3 Figure1]
Point 2: It is necessary to add in conclusion about the connections with sympathetic activity and the capacity to control lumbar movements (LMC). The negative correlation between LMC and LF/HF is not only a connection of proprioceptive mechanisms of movement control. An increase in the autonomic balance LF/HF indicates a strain of autonomic regulation due to stress associated with pain syndrome. It is this mechanism that weakens the patient's motor control.
Response 2:
We have added a statement of correlation to the conclusion. In addition, we have added a supplementary explanation in the Discussion section.
[P6 L224-235, P8 L315-317]
Reviewer 2 Report
Comments and Suggestions for Authors
Basic reporting
Dear authors, the manuscript is generally well-written and easy to read; a slight spell-check is required. I have just some concerns that the authors must address.
Abstract
keywords usually should be different from that used in the main title.
Introduction
The literature on the subject is sufficiently well summarised. However, it could be useful to clarify some information about:
- You define LBP as a general term for pain in the lumbar region, but you don’t distinguish between acute, subacute, and chronic LBP, which could have different causes, treatments, and outcomes. Maybe, this generalization might weaken the argument for creating effective subgroups.
- While subgrouping is presented as a promising approach, you don’t critically evaluate potential challenges or limitations (e.g., variability within subgroups, difficulty in applying findings clinically).
- There is no clear explanation of how previous studies have failed to use multidimensional data effectively or what gaps your study intends to fill.
Methods
- Figure 1 is not so clear. You mean that no one of the 154 subjects met any of the exclusion criteria?
- Maybe, some exclusion criteria, such as difficulty performing certain measurement positions, may inadvertently exclude participants with more severe chronic LBP, leading to an underrepresentation of such cases.
- Variations in participant signalling, during LPPT measurement, could introduce subjectivity.
- Shortened measurement times for HRV (1 minute instead of 3) could impact reliability, even if Bland-Altman analysis confirmed no systematic error.
- You should justify the statistical methods, including the sample size adequacy for cluster analysis since hierarchical cluster analysis with a small sample size (n = 46) may yield unstable clusters.
- You should also clearly state criteria for choosing between ANOVA and Kruskal-Wallis tests.
Validity of the findings
- The reported correlations (e.g., r=0.37) are statistically significant but relatively weak. This raises questions about their clinical relevance. You should discuss whether such small correlations have meaningful implications for understanding LBP-related parameters.
- Furthermore, the p-value (p=0.01) does not imply practical significance; the effect sizes and their real-world implications could be more useful.
- I could be wrong but, some groups (e.g., Clusters 1 and 2) share similar traits (e.g., low trunk muscle mass and LPPT). This overlap may indicate suboptimal cluster separation or inappropriate clustering variables.
- You report a negative correlation between LMC ability and autonomic balance, specifically stating that higher LMC ability is associated with higher sympathetic or lower parasympathetic activity. However, the mechanism behind this relationship is not fully explained. It not clear to me because you mention that when the sympathetic nervous system becomes dominant, muscle activity increases, co-contraction is prioritized, and proprioception decreases, leading to inhibited movement control. This seems to contrast with the idea that a high LMC ability (which suggests better movement control) is linked to higher sympathetic activity.
- You should clearly mention whether any potential confounding variables (e.g., physical activity, sleep quality, nutritional status) were controlled for during the analysis, especially given their potential impact on LBP.
Author Response
Thank you for pointing this out. We have revised it as follows.
Abstract
Point 1:
keywords usually should be different from that used in the main title.
Response 1:
We have corrected keywords as you suggested.
“Muscle mass; Movement control; Autonomic balance; Tenderness threshold; Central sensitization”
[P1 Keywords]
Introduction
Point 2:
You define LBP as a general term for pain in the lumbar region, but you don’t distinguish between acute, subacute, and chronic LBP, which could have different causes, treatments, and outcomes. Maybe, this generalization might weaken the argument for creating effective subgroups.
Response 2:
Thank you for your insightful comment.
You are correct that acute, subacute, and chronic LBP can have different causes, treatments, and outcomes. In our study, we focused specifically on chronic LBP in middle-aged and older women. This focus was outlined in the Methods section, but we understand the need to clarify it more explicitly in the introduction and discussion.
To strengthen our argument and avoid overgeneralization, we have revised the manuscript to clearly state that our findings and subgrouping approach apply specifically to chronic LBP.
[P2]
Point 3:
While subgrouping is presented as a promising approach, you don’t critically evaluate potential challenges or limitations (e.g., variability within subgroups, difficulty in applying findings clinically).
Response 3:
We have added a statement of caveats and challenges to the subgrouping in the Discussion section.
“Moreover, in this study, subgroups were identified based on key factors; however, individual variability within subgroups may affect the generalizability of the results. Therefore, further validation with larger sample sizes and evaluations of the reproducibility of clustering results are necessary to ensure the generalizability of the clusters for clinical application. Additionally, although subgrouping may streamline the evaluation process by reducing the number of assessment items, the complexity of evaluating the specific characteristics of each subgroup cannot be avoided. Furthermore, the lack of clear criteria for grouping highlights the need to establish evaluation protocols, including reference values, and to reconsider variables that can distinctly separate groups for future clinical application of these findings.”
[P7 L278-287]
Point 4:
There is no clear explanation of how previous studies have failed to use multidimensional data effectively or what gaps your study intends to fill.
Response 4:
Thank you for pointing out this important aspect. We recognize the need to provide a clearer explanation of how previous studies have approached multidimensional data and the specific gaps our study aims to address. To address this, we have added the following statement to the 'Introduction' section.
“The limited use of multidimensional data in previous studies may be attributed to the diversity of factors associated with CLBP and complexity of their interactions [19]. Focusing on clinically important and modifiable factors and establishing multidimensional sub-groups is expected to contribute to the development of novel classifications that enhance the effectiveness of interventions for middle-aged and older adults with CLBP.”
[P2 L76-80]
Point 5:
Figure 1 is not so clear. You mean that no one of the 154 subjects met any of the exclusion criteria?
Response 5:
We have corrected it as you suggested.
[P3 Figure1]
Point 6:
Maybe, some exclusion criteria, such as difficulty performing certain measurement positions, may inadvertently exclude participants with more severe chronic LBP, leading to an underrepresentation of such cases.
Response 6:
We have added the following statement to the study limitations in the Discussion section.
“This limitation is also reflected in the exclusion of participants in this study who had difficulty assuming the required measurement position.”
[P8 L292-294]
Point 7:
Variations in participant signalling, during LPPT measurement, could introduce subjectivity.
Response 7:
We clearly stated the signaling instructions given to participants.
“The LPPT was measured by instructing participants to say ‘stop’ when the pressure sen-sation changed to “uncomfortable” or “painful”; the average of three measurements was taken as the representative value [26].”
[P4 L143-144]
Point 8:
Shortened measurement times for HRV (1 minute instead of 3) could impact reliability, even if Bland-Altman analysis confirmed no systematic error.
Response 8:
As you pointed out, the potential impact of shortening the measurement duration for autonomic nervous system evaluation cannot be entirely excluded. Therefore, we have added the following statement to the limitations section of the study.
“In evaluating autonomic balance, this study adopted a 1-min measurement after con-firming the absence of systematic error using the Bland−Altman analysis. However, de-spite differences in devices and measurement sites, the measurement duration is shorter than the 3-min duration recommended in previous studies [25], and the potential effect of this shortened duration on the results cannot be entirely excluded. Additionally, although this study utilized the LF/HF ratio as an indicator of autonomic balance, future studies should consider incorporating other parameters, such as absolute values like total power [25], which may require longer measurement durations for accurate evaluation.”
[P8 L300-308]
Point 9:
You should justify the statistical methods, including the sample size adequacy for cluster analysis since hierarchical cluster analysis with a small sample size (n = 46) may yield unstable clusters.
Response 9:
Thank you for pointing out this important aspect.
We acknowledge that the small sample size and the potential for reduced stability are limitations of our study. To address this concern, we have included the following points in the manuscript:
- Applicability of hierarchical cluster analysis
Hierarchical cluster analysis is widely used for exploratory data analysis and is considered suitable for small to moderate sample sizes (e.g., Everitt et al., 2011).
[P4 L167-168]
- Validation of cluster validity
To support the validity of the identified clusters, additional statistical comparisons between clusters were conducted. The results have been included in the manuscript to demonstrate that the clustering reflects the actual data structure.
[P4 L170-171]
- Discussion on sample size
In the Discussion section, we addressed the limitations associated with the small sample size and its potential impact on the study results. We also emphasized the need for further validation using larger sample sizes as a direction for future research.
[P7 L280-282]
Point 10:
You should also clearly state criteria for choosing between ANOVA and Kruskal-Wallis tests.
Response 10:
We have added to the Methods section that the selection criteria were based on normality, and we have also included which variables did not meet normality in the Results section.
[P4-5 L171-174, P5 L178]
Validity of the findings
Point 11:
The reported correlations (e.g., r=0.37) are statistically significant but relatively weak. This raises questions about their clinical relevance. You should discuss whether such small correlations have meaningful implications for understanding LBP-related parameters.
Response 11:
We have added the following statement regarding interpretation based on the magnitude of the correlation coefficient.
“In this study, significant correlations were identified between the LMC test score and autonomic balance (r = −0.37) and between trunk muscle mass and LPPT (r = 0.37), both demonstrating a moderate magnitude [33]. These results provide important preliminary insights into the potential relationships among multidimensional parameters associated with CLBP. The observed correlation magnitudes suggest that CLBP is not solely influenced by a single factor, but rather by the combined effects of multiple interacting factors.”
[P6 L216-221]
Point 12:
Furthermore, the p-value (p=0.01) does not imply practical significance; the effect sizes and their real-world implications could be more useful.
Response 12:
We added something about AE in the Results and Discussion section because AE was confirmed to be a large effect size with no significant difference.
[P6 L204-206, P7 L257-261, P7 L275-276]
Point 13:
I could be wrong but, some groups (e.g., Clusters 1 and 2) share similar traits (e.g., low trunk muscle mass and LPPT). This overlap may indicate suboptimal cluster separation or inappropriate clustering variables.
Response 13:
The fact that Cluster 1 and Cluster 2 share similar characteristics (e.g., trunk muscle mass and LPPT) may be attributed to the nature of hierarchical clustering, where the selection of dendrogram cutoff points can influence the resulting cluster structure. While such overlaps are to some extent unavoidable in hierarchical clustering, the separation between clusters could be further clarified by reconsidering the variables used, taking into account correlations between them.
Specifically, adding new relevant variables or incorporating variables with lower correlations may help to better distinguish the characteristics of each cluster. This point has been included in the discussion section of the manuscript.
[P7 L278-282]
Point 14:
You report a negative correlation between LMC ability and autonomic balance, specifically stating that higher LMC ability is associated with higher sympathetic or lower parasympathetic activity. However, the mechanism behind this relationship is not fully explained. It not clear to me because you mention that when the sympathetic nervous system becomes dominant, muscle activity increases, co-contraction is prioritized, and proprioception decreases, leading to inhibited movement control. This seems to contrast with the idea that a high LMC ability (which suggests better movement control) is linked to higher sympathetic activity.
Response 14:
As you pointed out, the description was difficult to understand and the intent was not clear, therefore, we have corrected it.
[P6 L223-235]
Point 15:
You should clearly mention whether any potential confounding variables (e.g., physical activity, sleep quality, nutritional status) were controlled for during the analysis, especially given their potential impact on LBP.
Response 15:
As you indicated, lifestyle-related factors were not investigated, and we have added this to the study limitations.
[P8 L308-311]
Round 2
Reviewer 2 Report
Comments and Suggestions for Authors
the atuhors adressed all my conserns. I have no further suggestions.